Histological variability in the limb bones of the Asiatic wild ass and its significance for life history inferences

Nacarino-Meneses Carmen carmen.nacarino@icp.cat 1
Jordana Xavier 1
Köhler Meike 1 2 3
1 Department of Evolutionary Biology, Institut Català de Paleontologia Miquel Crusafont (ICP) , Campus de la Universitat Autònoma de Barcelona, Bellaterra, Barcelona , Spain
2 Institut Català de Recerca i Estudis Avançats (ICREA) , Barcelona , Spain
3 Department of Animal Biology, Plant Biology and Ecology (BABVE), Universitat Autònoma de Barcelona , Bellaterra, Barcelona , Spain
Jungers William
Electronic publication date: 2016 Oct 13
Publication date: 2016
Volume: 4
Electronic Location ID: e2580
Received 2016 Jul 5; Accepted 2016 Sep 18
Copyright: ©2016 Nacarino-Meneses et al.
Copyright year: 2016
Copyright holder: Nacarino-Meneses et al.
License: This is an open access article distributed under the terms of the Creative Commons Attribution License, which permits unrestricted use, distribution, reproduction and adaptation in any medium and for any purpose provided that it is properly attributed. For attribution, the original author(s), title, publication source (PeerJ) and either DOI or URL of the article must be cited.
License URL: https://creativecommons.org/licenses/by/4.0/

Keywords: Equus hemionus, Bone histology, External fundamental system, Bone growth marks, Life history, Skeletochronology, Intraskeletal histological variability, Longevity, Reproductive maturity, Limb bones

Funding: Spanish Ministry of Economy and Competitiveness (MINECO) CGL2015-63777 The Government of Catalonia 2014-SGR-1207 MINECO BES-2013-066335 This work is supported by the Spanish Ministry of Economy and Competitiveness (MINECO) (PI: M.K. & X.J., CGL2015-63777) and by the Government of Catalonia (2014-SGR-1207). C.N.-M. holds a grant from the MINECO (BES-2013-066335). The funders had no role in study design, data collection and analysis, decision to publish, or preparation of the manuscript.

==============================
The study of bone growth marks (BGMs) and other histological traits of bone tissue provides insights into the life history of present and past organisms. Important life history traits like longevity or age at maturity, which could be inferred from the analysis of these features, form the basis for estimations of demographic parameters that are essential in ecological and evolutionary studies of vertebrates. Here, we study the intraskeletal histological variability in an ontogenetic series of Asiatic wild ass (Equus hemionus) in order to assess the suitability of several skeletal elements to reconstruct the life history strategy of the species. Bone tissue types, vascular canal orientation and BGMs have been analyzed in 35 cross-sections of femur, tibia and metapodial bones of 9 individuals of different sexes, ages and habitats. Our results show that the number of BGMs recorded by the different limb bones varies within the same specimen. Our study supports that the femur is the most reliable bone for skeletochronology, as already suggested. Our findings also challenge traditional beliefs with regard to the meaning of deposition of the external fundamental system (EFS). In the Asiatic wild ass, this bone tissue is deposited some time after skeletal maturity and, in the case of the femora, coinciding with the reproductive maturity of the species. The results obtained from this research are not only relevant for future studies in fossil Equus, but could also contribute to improve the conservation strategies of threatened equid species.

Introduction

The study of bone growth marks (BGMs) is nowadays the focus of many investigations due to its potential to reconstruct many aspects of the life history of present and past vertebrates (Amson et al., 2015; Kolb et al., 2015a; Woodward et al., 2015; Jordana et al., 2016; Moncunill-Solé et al., 2016; Nacarino-Meneses, Jordana & Köhler, 2016; Orlandi-Oliveras et al., 2016). These histological features, which record cyclic variation in bone growth rate, can take the form of “lines of arrested growth” (LAGs) or of “annuli” within the cortical bone (Castanet et al., 1993). LAGs appear as thin dark lines in bone cross-sections and are considered to represent moments of cessation of growth (Francillon-Vieillot et al., 1990; Chinsamy-Turan, 2005). Annuli, on the other hand, are poorly vascularized rings of lamellar or parallel-fibered bone within the bone cortex (Francillon-Vieillot et al., 1990; Chinsamy-Turan, 2005) that indicate periods of growth rate decrease. From Peabody (1961) to the present, it has repeatedly been demonstrated that most of the BGMs found in the bone tissue record annual cycles of growth (cyclical growth marks—CGMs) reflecting physiological cycles (Köhler et al., 2012) that match environmental cycles (Castanet et al., 1993; Chinsamy-Turan, 2005). Nevertheless, BGMs are also suggested to register biological events that entail moments of physiological stress in the organism (Woodward, Padian & Lee, 2013) instead of periodical growth (Castanet, 2006).

From dinosaurs to mammals, the annual periodicity of the CGMs is the basis for inferences of life history strategies in many groups of fossil organisms (e.g., Klevezal, 1996; Horner, De Ricqlès & Padian, 2000; Köhler & Moyà-Solà, 2009). The number of CGMs within a bone cortex allows researchers to calculate important life history traits such as longevity (Castanet et al., 2004; Köhler & Moyà-Solà, 2009; Köhler, 2010) or age at maturity (Chinsamy & Valenzuela, 2008; Horner, De Ricqlès & Padian, 2000; Köhler & Moyà-Solà, 2009; Köhler, 2010; Marín-Moratalla, Jordana & Köhler, 2013; Jordana et al., 2016) by means of a technique called skeletochronology (Castanet et al., 1993). This method also provides information about other biological aspects of the animals such as their growth strategy or physiology (Horner, De Ricqlès & Padian, 2000; Padian, De Ricqlès & Horner, 2001; Köhler et al., 2012; Woodward et al., 2015). However, skeletochronology has some limitations that are particularly important when dealing with mammals. Firstly, the remodelling process (haversian systems) and the expansion of the medullary cavity that accompany the increase in age can hide the presence of previous CGMs and, thus, give an underestimated individual age (Woodward, Padian & Lee, 2013). The inference of this important trait could also be altered if non-cyclical BGMs are erroneously counted as cyclical ones. On the other hand, CGMs are difficult to identify if they are located in the lamellar and avascular bone tissue deposited in the outermost cortex of adult individuals (external fundamental system—EFS) (Woodward, Padian & Lee, 2013), because of the structural similarity between LAGs and the lamellae of this tissue (Horner, De Ricqlès & Padian, 1999). Such misidentification of CGMs within the EFS, along with the fact that mammals present asymptotic growth (Lee et al., 2013), also reduces the accuracy of longevity estimates when old specimens are analyzed (Castanet et al., 2004; Woodward, Padian & Lee, 2013). Finally, several authors had reported a variable number of CGMs depending on the bone analyzed within an individual (García-Martínez et al., 2011; Woodward, Horner & Farlow, 2014). Thus, it is important to select the most appropriate bone for skeletochronological studies in each taxon before making general assessments about the life history of the species (Horner, De Ricqlès & Padian, 1999).

The histological analysis of bones for this kind of research in mammals is still little explored in comparison with other vertebrate groups (Castanet et al., 2004; Kolb et al., 2015a; Jordana et al., 2016). However, since the study of Köhler et al. (2012) that demonstrated the correlation between cyclical bone growth and seasonal physiology in a wide sample of ruminants, the number of histological works in extant (Marín-Moratalla, Jordana & Köhler, 2013; Marín-Moratalla et al., 2014; Jordana et al., 2016; Nacarino-Meneses, Jordana & Köhler, 2016) and extinct mammals (Martínez-Maza et al., 2014; Kolb et al., 2015b; Amson et al., 2015; Moncunill-Solé et al., 2016; Orlandi-Oliveras et al., 2016) has considerably increased. Among all mammalian clades, members of the family Equidae play a key role in extant and fossil ecosystems (MacFadden, 1992; Downer, 2014). Besides, they are a classical group of research in Paleontology due to their characteristic evolution (MacFadden, 2005). Nevertheless, histological studies in equids are scarce and only a few aimed to infer the life history strategies of some fossil (Sander & Andrássy, 2006; Martínez-Maza et al., 2014) or extant representatives (Nacarino-Meneses, Jordana & Köhler, 2016) of the group.

For the reasons set out above, the main objective of the present work is to study the histological variability (BGMs, pattern of vascularization, bone tissue types) between different limb bones of the same individual in the Asiatic wild ass (Equus hemionus Pallas, 1775). With this study, we aim to find out what life history information can be inferred from the histological study of equids and to try to determine which is the best skeletal element to develop skeletochronological studies in this mammal. The kulan or Asiatic wild ass, a mammal endemic to the Gobi desert, is one of the eight extant species of the family Equidae (Steiner & Ryder, 2011) and presents nowadays a delicate conservation status (Kaczensky et al., 2015). Because previous studies pointed out the potential of histological analyses in conservation management of wild populations (Chinsamy & Valenzuela, 2008; García-Martínez et al., 2011; Marín-Moratalla, Jordana & Köhler, 2013), we have considered this species as the most appropriate to conduct this study. Moreover, its extant habitat—the steppe and semi-desert plains of Mongolia, Iran, Turmekistan, India and China (Feh et al., 2001; Reading et al., 2001; Kaczensky et al., 2015)—make this extant taxon the most similar to fossil stenoid horses (Forstén, 1992) extending the importance of our research form Conservation Biology to Palaeontology.

Material and Methods

Thin sections from femur, tibia, metatarsus and metacarpus were analyzed in an ontogenetic series of 9 specimens of E. hemionus (Table 1). Only specimen IPS83154 lacks metacarpal bone, totaling 35 the cross-sections studied. As shown in Table 1, the sample includes individuals from different habitats, sex and ages. Sex data were provided by curators while age at death was estimated according to dental eruption pattern of the species (Lkhagvasuren et al., 2013) and corroborated with the analysis of cementum layers in adult individuals (R Schafberg, pers. comm., 2014). Wild specimens (IPS83876–IPS83877) were collected during the Mongolian-German Biological Expeditions in the Gobi desert (Schöpke et al., 2012) and are housed at the Natural History Collections of the Martin-Luther-University Halle-Wittenberg (Halle, Germany). Captive individuals (IPS83149–IPS83155) lived in the Hagenbeck Zoo (Hamburg, Germany) and belong to the collections of the Zoological Institute of Hamburg University (Hamburg, Germany).

Table 1 Sample studied.

Individual	Estimated age	Age group	Habitat	Sex	Bones studied	Collection	
IPS83152	<3 weeks	Perinatal	Hagenbeck Zoo	−	Fe, Ti, Mc, Mt	Zoological Institute of Hamburg University (Hamburg, Germany)	
IPS83153	0.5 years	Foal	Hagenbeck Zoo	M	Fe, Ti, Mc, Mt	Zoological Institute of Hamburg University (Hamburg, Germany)	
IPS83154	0.5 years	Foal	Hagenbeck Zoo	M	Fe, Ti, Mc	Zoological Institute of Hamburg University (Hamburg, Germany)	
IPS83149	1 year	Yearling	Hagenbeck Zoo	−	Fe, Ti, Mc, Mt	Zoological Institute of Hamburg University (Hamburg, Germany)	
IPS83150	1 year	Yearling	Hagenbeck Zoo	−	Fe, Ti, Mc, Mt	Zoological Institute of Hamburg University (Hamburg, Germany)	
IPS83151	1 year	Yearling	Hagenbeck Zoo	−	Fe, Ti, Mc, Mt	Zoological Institute of Hamburg University (Hamburg, Germany)	
IPS83155	2 years	Juvenile	Hagenbeck Zoo	F	Fe, Ti, Mc, Mt	Zoological Institute of Hamburg University (Hamburg, Germany)	
IPS83876	4.5 years	Adult	Gobi desert	F	Fe, Ti, Mc, Mt	Museum of Domesticated Animals (Halle, Germany)	
IPS83877	8 years	Adult	Gobi desert	M	Fe, Ti, Mc, Mt	Museum of Domesticated Animals (Halle, Germany)	
Notes.

M male

F female

Fe femur

Ti tibia

Mc metacarpus

Mt metatarsus

From the mid-shaft of each bone, we prepared histological slices following standard procedures in our laboratory (Nacarino-Meneses, Jordana & Köhler, 2016). After measuring and photographing each bone, three centimeters of its mid-shaft were cut and embedded in an epoxy resin (Araldite 2020). This block was later cut into two halves (ISO Met, Biometa) and the exposed surface was polished with carborundum powder to be fixed to a frosted glass with an UV curing glue (Loctite 358). Afterwards, it was cut with a diamond saw (Petrothin, Buehler) up to a thickness of 100–120 microns and polished again with carborundum powder. Finally, a mix of oils (Lamm, 2013) was spread over the slice before being sheltered with a cover slip. Longitudinal sections were also prepared from several blocks to corroborate that the identification of bone tissue types does not rely on the orientation of the cutting plane (Stein & Prondvai, 2014). All thin-sections were observed in a Leica DM 2500P microscope under polarized light with a 1/4λ filter and photographed with the camera incorporated in the microscope. The use of a retardation filter that colors the cross-section, which is not mandatory in this kind of studies, was used to improve the visualization of BGMs and to facilitate the description of bone histology and skeletochronology (Turner-Walker & Mays, 2008).

To analyze the histological variability between skeletal elements, bone tissue types and BGMs were studied. The histological descriptions follow the classification of Francillon-Vieillot et al. (1990) and De Margerie, Cubo & Castanet (2002). The terminology proposed by Prondvai et al. (2014) was employed to describe the different components of the fibrolamellar complex (FLC) (a special case of woven-parallel complex for this authors): “fibrous” or woven bone (WB) and “lamellar” or parallel-fibered bone (PFB). Because the femoral bone histology of the Asiatic wild ass has previously been described in detail (Nacarino-Meneses, Jordana & Köhler, 2016), only descriptions of the bone tissue of tibiae, metacarpi and metatarsi will be detailed in the present work. Regarding growth marks, we have generally used the term “bone growth mark—BGM,” interchangeably for LAGs or annuli, instead of “cyclical growth mark—CGM” because not all the marks identified in the samples have proved to be periodical. Double LAGs or LAGs that split were considered as a single event. BGMs were traced along the cross-sections and superimposition of individuals was performed to identify growth marks that have been erased by the remodeling process or the expansion of the medullary cavity (Woodward, Padian & Lee, 2013). Each BGM circumference was measured with ImageJ® software to estimate the bones’ perimeter at different times during ontogeny and the results were plotted to obtain growth curves for each sample (Bybee, Lee & Lamm, 2006). The perimeter of the cross-section was also calculated with ImageJ® software in those animals that are still growing (subadult individuals) to estimate its bone perimeter at the time of death. The perimeter of adult individuals was not determined and only the length of the BGMs identified within the EFS is shown. Because it is generally considered that the presence of EFS indicates the cessation of radial growth in long bones (Huttenlocker, Woodward & Hall, 2013), the length of the BGMs located in this bone tissue and the perimeter of the cross-section are almost the same value. Thus, the estimation of the cross-section’s perimeter in adult specimens does not provide relevant information about the growth of the animal. Furthermore, we calculated the size variation per year of each bone in yearling and adult specimens as the difference of BGMs’ perimeters of consecutive annual growth cycles and interpreted it as a proxy of growth rate. Finally, several life history traits were calculated in each bone from the study of CGMs. Age at death of the specimens was determined as the total number of CGMs present in the bone cortex (Castanet et al., 2004) and compared with the age estimated from teeth. Age at maturity was calculated by counting the CGMs before the deposition of the EFS (Chinsamy & Valenzuela, 2008; Marín-Moratalla, Jordana & Köhler, 2013) and contrasted with literature data.

Results

Bone tissue types

All bones of E. hemionus present a well-vascularized FLC that is progressively remodeled during ontogeny. However, the arrangement of the vascular canals embedded in the FLC varies among the bones sampled and in the course of ontogeny. An ontogenetic change in the proportion of the different components of the bone matrix (WB and PFB) has also been noted in some of the limb bones studied, regardless of the orientation of the cutting plane (transversal or longitudinal preparations).

The histology of kulan’s femora was previously described in Nacarino-Meneses, Jordana & Köhler (2016). It consists of a highly vascularized FLC that presents an ontogenetic change in the orientation of the vascular canals to a predominantly circumferential arrangement, along with a decrease in the proportion of the WB of the matrix. The EFS was only indentified in adult stages and remodeling was associated to the course of ontogeny and to mechanical loading.

Tibial cortices consist of laminar bone (Fig. 1A) and remodeling begins early in ontogeny, as the high number of secondary osteons (SO) identified in yearling specimens (Fig. 1B) suggests. Regarding primary bone tissue, the cortical bone of the perinatal individual presents FLC with a high proportion of PFB in the bone matrix (Fig. 1C). The cortex of foals, as well as those of yearling and juvenile individuals, is divided into two well-defined areas that differ in the proportion of this bone matrix component. In these specimens, the laminar bone of the internal cortex presents a higher proportion of PFB than the outer one (Fig. 1A). The EFS is not identified in any of the tibiae analyzed. Instead, several packages of a poorly vascularized lamellar bone that interrupt the FLC matrix, can be recognized in the mid-outer cortex of adult specimens (Fig. 1D). This bone tissue differs from the real EFS (Huttenlocker, Woodward & Hall, 2013) because it is not restricted to the outermost cortex.

Figure 1 Tibial bone histology of the Asiatic wild ass.

(A) Detail of the lateral cortex of the foal IPS83153, showing two areas that differ in the proportions of the parallel-fibered component (PFB) of the bone matrix. (B) Haversian systems in the anterior cortex of the yearling IPS83150. (C) Anterior cortex of the newborn individual (IPS83152) with a high proportion of parallel-fibered component (PFB) in its bone matrix. (D) Packages of lamellar bone within the fibrolamellar complex in the anterior cortex of the wild male (IPS83877). HS, haversian systems; LB, lamellar bone; PFB, parallel-fibered bone. Scale bars: 1 millimeter. All images were obtained under polarized light with a 1/4λ filter.

Bone tissue and vascular arrangement is very similar in metatarsi and metacarpi. In both skeletal elements, the bone cortex is mainly composed of a FLC with primary osteons (POs) oriented in circular rows (Fig. 2A). The vascular canals of these POs present a larger diameter in the outer half of the cortex than in the inner half (Fig. 2A). Some radial canals are situated in the proximity of the medullary cavity in metacarpal bones (Fig. 2B) whereas metatarsi present several areas with laminar bone (Fig. 2C). Haversian bone is restricted to the posterior side of the cortex in immature kulans but it is more generalized in adult ones. The EFS is identified in the outermost cortex of adult individuals (Fig. 2D).

Figure 2 Metapodial bone histology of the Asiatic wild ass.

(A) Anterior metatarsal cortex of the yearling IPS83149, showing a fibrolamellar complex with primary osteons oriented in circular rows. (B) Radial canals in the metacarpus of the yearling IPS83150. (C) Circular canals in the metatarsus of the foal IPS83153. (D) Detail of the external fundamental system in the metatarsus of the wild female IPS83876. EFS, external fundamental system; FLC, fibrolamellar complex. Scale bars: 1 millimeter. All images were obtained under polarized light with a 1/4λ filter.

Bone growth marks

Table 2 shows the number of BGMs identified in the different bones of each individual. From foals to adults, all samples present these features, although its number varies among skeletal elements of the same individual and between individuals of the same age category.

Table 2 Number of bone growth marks (BGMs) identified in each cross-section.

				Femur	Tibia	Metacarpus	Metatarsus	
Individual	Estimated age	Age group	Sex	FLC	EFS	Total	FLC	EFS	Total	FLC	EFS	Total	FLC	EFS	Total	
IPS83152	<3 weeks	Perinatal	−	0	−	0	0	−	0	0	−	0	0	−	0	
IPS83153	0.5 years	Foal	M	0	−	0	1	−	1*	1		1*	1	−	1*	
IPS83154	0.5 years	Foal	M	0	−	0	1	−	1*	1	−	1*	−	−	−	
IPS83149	1 year	Yearling	−	2	−	2*	2	−	2*	2	−	2*	2	−	2*	
IPS83150	1 year	Yearling	−	2	−	2*	2	−	2*	2	−	2*	2	−	2*	
IPS83151	1 year	Yearling	−	1	−	1*	1	−	1*	1	−	1*	1	−	1*	
IPS83155	2 years	Juvenile	F	1	−	1	2	−	2*	1	−	1	2	−	2*	
IPS83876	4.5 years	Adult	F	4	1	5	4	−	4	3	2	5*	3	2	5*	
IPS83877	8 years	Adult	M	4	2	6	5	−	5	4	2	6*	4	2	6*	
Notes.

M male

F female

FLC number of BGMs identified within the fibrolamellar complex

EFS number of BGMs identified within the external fundamental system

* Indicates that the most internal BGM has been considered as a non-cyclical BGM.

The presence of a BGM in the middle cortex of tibia, metacarpus and metatarsus (Fig. 3, Table 2) of foals (IPS83153 and IPS83154) is surprising. LAGs and annuli are known to be annual and deposited during the unfavorable season (i.e., winter for E. hemionus) in mammals (Köhler et al., 2012). Because kulans tend to give birth in summer (Zuckerman, 1952; Nowak, 1999; Feh et al., 2001; Feh et al., 2002) and our foals are around six months old (Table 1), the CGM corresponding to the first winter should be observed in the outermost cortex, not in the mid-cortex (Fig. 3). Therefore, this feature is interpreted as a non-cyclical growth mark and will not be taken into account for age estimation.

Figure 3 Bone growth marks in foal kulans.

(A) BGM in the lateral side of the tibia (IPS83154). (B) BGM in the anterior cortex of the metacarpus (IPS83153). (C) BGM in the anterior side of the metatarsus (IPS83153). White arrows indicate bone growth marks. Scale bar: 1 millimeter. All images were obtained under polarized light with a 1/4λ filter.

Yearling specimens (IPS83149, IPS83150 and IPS83151) present a variable number of LAGs. As it is shown in Table 2, one BGM is identified in all skeletal elements of IPS83151, while IPS83149 and IPS83150 present two (Fig. 4). Such variability might be explained by the fact that the first permanent molar is totally unworn in IPS38151 but presents initial wear in IPS83149 and IPS83150. Thus, the former might be somewhat younger than the others. Because these specimens are aged as one year, we interpret the most external BGM identified in all bones of IPS83149 and IPS83150 (Figs. 4B and 4D) as CGM deposited during the first year of life. However, we consider the internal BGM observed in these individuals (Figs. 4B and 4D), as well as the single BGM identified in the mid-cortex of all bones of IPS83151 (Figs. 4A and 4C), as a non-cyclical growth mark.

Figure 4 Bone growth marks in yearling kulans.

(A) Femoral bone cortex of IPS83151 showing one BGM in its anterior side. (B) Tibial bone cortex of IPS83150 showing two BGMs in its lateral side. (C) Metacarpal bone cortex of IPS83151 showing one BGM in its lateral side. (D) Metatarsal bone cortex of IPS83149 showing two BGMs in its anterior side. White arrows indicate bone growth marks. Scale bar: 1 millimeter. All images were obtained under polarized light with a 1/4λ filter.

Two BGMs are identified in the tibia and the metatarsus of the juvenile individual (IPS83155) while the femur and the metacarpus present only one (Table 2, Fig. 5). In these latter bones, the growth mark appears in the outer cortex (Figs. 5A and 5C). Because this individual is aged around two years, we consider that this external BGM is representing the winter growth arrest during its second year of life. The second BGM in the tibia and metatarsus is also found in the external part of the cortex (Figs. 5B and 5D), so we interpret it as the CGM corresponding to the second winter. On the other hand, superimposition of individuals reveals that the first BGM of these bones (Figs. 5B and 5D) does not correspond to the CGM identified in yearlings, as it appears more internally within the cortex. This fact suggests that the first winter has not been recorded in this animal and that such internal BGM could be considered as non-cyclical.

Figure 5 Bone growth marks in the juvenile kulan (IPS83155).

(A) Femoral bone cortex showing one BGM in its anterior side. (B) Tibial bone cortex showing two BGMs in its lateral side. (C) Metacarpal bone cortex showing one BGM in its anterior side. (D) Metatarsal bone cortex showing two BGMs in its anterior side. White arrows indicate bone growth marks. Scale bar: 1 millimeter. All images were obtained under polarized light with a 1/4λ filter.

Wild adult individuals (IPS83876 and IPS83877) also present differences in the number of BGMs between limb bones (Table 2, Fig. 6). Femur, metatarsus and metacarpus of the wild female (IPS83876) show five BGMs while only four BGMs are identified in its tibia (Table 2, Figs. 6A, 6B, 6E and 6F). In the femur, four BGMs lie within the FLC and one within the avascular and highly organized lamellar tissue deposited in the periphery of the bone (EFS) (Table 2, Figs. 6A and 6B). Metapodial bones, however, present three BGMs within the FLC and two BGMs in the EFS (Table 2, Figs. 6E and 6F). The four BGMs found in the tibia are within the FLC, as an EFS is not identified in this bone. On the other hand, the wild male (IPS83877) presents six BGMs in its femur and metapodial bones, whereas five BGMs are found in the tibia (Table 2, Figs. 6C, 6D, 6G and 6H). Superimposition of both adult individuals reveals that one BGM has been lost in the femur of the wild male due to bone remodeling (Nacarino-Meneses, Jordana & Köhler, 2016). This process, however, has not erased the presence of any BGM in the other limb bones studied. Thus, a total of seven BGMs should be counted in the femur of the wild male: five in the FLC (one hidden by secondary osteons) and two in the EFS. Five BGMs, all located in the FLC, are identified in the tibia of this wild male (IPS83877; Table 2, Figs. 6C and 6D). Finally, four BGMs are found in the FLC and two in the EFS of its metatarsus and metacarpus (Table 2, Figs. 6G and 6H). The correspondence between the age of both adults and the number of BGMs identified in their limb bones indicates that all these features could be considered as CGMs. However, superimposition suggests that the most internal BGM observed in metapodial bones of wild adults might be a non-cyclical feature, as they are deposited previously to the CGM identified in yearlings.

Figure 6 Bone growth marks in adult kulans.

(A) Femoral bone cortex of the wild female (IPS83876) showing five BGMs in its anterior side. (B) Detail of the most external BGMs identified in the femur of IPS83876. Fifth BGM is located within the external fundamental system. (C) Tibial bone cortex of the wild male (IPS83877) showing five BGMs in its lateral side. (D) Detail of the most external BGMs identified in the tibia of IPS83877. (E) Metacarpal bone cortex of the wild female (IPS83876) showing five BGMs in its anterior side. (F) Detail of the most external BGMs identified in the metacarpus of IPS83876. Fourth and fifth BGMs are located within the external fundamental system. (G) Metatarsal bone cortex of the wild male (IPS83877) showing six BGMs in its anterior side. (H) Detail of the most external BGMs identified in the metacarpus of IPS83877. Fifth and sixth BGMs are located within the external fundamental system. White dashed rectangles indicate areas of image magnifications. White arrows indicate bone growth marks. White scale bar: 1 millimeter; black scale bar: 500 microns. All images were obtained under polarized light with a 1/4λ filter.

Growth curves

Based on the ontogenetic time schedule obtained from the study of the BGMs, we represented the growth curve for the different bones of each specimen (Figs. 7A–7D). In these graphs, the perimeter of the bone (outline of the BGM) at different years is plotted against the estimated age. Because the non-cyclical BGM identified in several bones is deposited sometime before the six months of life (Table 2, Fig. 3), it has been considered as time “zero” in the growth curves. The amount of growth in successive years, calculated as a proxy of growth rate, is also represented for yearlings and adult kulans (Figs. 7E–7H).

Figure 7 Bone growth of the Asiatic wild ass.

From (A–D), bone perimeter (mm, ordinate axis) is plotted against estimated age (years, abscissa axis) to obtain growth curves. From (E–F), variation of bone perimeter (mm, ordinate axis) is plotted against estimated age (years, abscissa axis) as a proxy of growth rate. (A) Growth curves obtained from the femora. (B) Growth curves obtained from the tibiae. (C) Growth curves obtained from the metacarpi. (D) Growth curves obtained from the metatarsi. (E) Femoral growth rate. (F) Tibial growth rate. (G) Metacarpal growth rate. (H) Metatarsal growth rate. Legend is shown in the bottom of the figure. In the graphs, filled characters represent females, unfilled ones correspond to males and linear ones indicate animals with unknown sex. Dashed lines indicate wild animals while continuous lines represent captive ones. Male and female symbols indicate the time of deposition of the external fundamental system (EFS) in each wild adult respectively. It could be noted that this moment does not match with the decline in periosteal growth rate.

In adult individuals, the growth curves, as well as the plots of growth rate estimations, indicate a change in the pace of growth during ontogeny. Figure 7A shows that in both adults, growth of the femur slows down at the fourth year of life and from this time onwards growth is minimal (Fig. 7E). However, this decrease in growth rate takes place at the age of two in tibia, metatarsus and metacarpus (Figs. 7B–7D), followed by only minimal growth (Figs. 7F–7H). Figure 7 also reveals differences in growth between captive and wild kulans. The results obtained from the analysis of bone growth cycles of the femur indicate two different growth tendencies with wild specimens growing more slowly than captives (Fig. 7A). While this difference is not perceived in the growth curves of the other limb bones studied (Figs. 7B–7D), growth rates of captive individuals are always higher than those of wild kulans in the first year of life (Figs. 7E–7H).

Discussion

In the present research, we analyzed the histological variability between limb bones in the extant species Equus hemionus for the first time. Previous studies have addressed this issue in isolated bones of fossil vertebrate species (Horner, De Ricqlès & Padian, 2000; Sander & Andrássy, 2006; Cullen et al., 2014; Martínez-Maza et al., 2014), but only a few have studied the histological variation of bone tissue within the same individual (Horner, De Ricqlès & Padian, 1999; García-Martínez et al., 2011; Woodward, Horner & Farlow, 2014; Cambra-Moo et al., 2015). Our analysis of kulan’s bone histology contributes to the knowledge of intraskeletal variability in mammals, providing new and important results that are of interest in different scientific areas. The applicability of histological studies to describe the life history of past animals and their evolutionary trends is well known (Köhler & Moyà-Solà, 2009; Marín-Moratalla et al., 2011; Martínez-Maza et al., 2014; Woodward et al., 2015). However, many researchers claim that more studies in living taxa are needed to truly understand the correlation between bone histology and the life history strategy of past organisms (Martínez-Maza et al., 2014; Woodward, Horner & Farlow, 2014; Cambra-Moo et al., 2015; Kolb et al., 2015a; Jordana et al., 2016). The results obtained from the present research will serve as a basis for the inference of life history parameters from the histology of extinct mammal species. Even more, skeletochronological studies of extant species are also of interest in related biological disciplines like Conservation Biology (Chinsamy & Valenzuela, 2008; García-Martínez et al., 2011; Marín-Moratalla, Jordana & Köhler, 2013). Nowadays, most of the wild species of the genus Equus are threatened and conservation policies are usually focus on genetic studies of captive individuals (Orlando, 2015). By means of skeletochronology, however, key life history traits such as longevity or age at sexual maturity can be inferred from the bone tissue of wild specimens (Castanet et al., 2004; Marín-Moratalla, Jordana & Köhler, 2013; Jordana et al., 2016). This information can be later used to calculate demographic parameters (e.g., life expectancy, generation time) that are essential to improve the conservation status of the species in the wild (Feh et al., 2001).

The detailed analysis of LAGs and annuli performed in the present research reveals that the number of BGMs recorded by the different limb bones varies within the same specimen (Table 2), a fact that has previously been reported for other vertebrate groups (Horner, De Ricqlès & Padian, 1999; García-Martínez et al., 2011; Cullen et al., 2014; Woodward, Horner & Farlow, 2014). Our results show that the femur registers the highest total number of BGMs, as well as the highest number of these features within the FLC (Table 2). This observation, which has previously been observed in mammals (García-Martínez et al., 2011), is likely related with the fact that the femur is the bone that more tightly correlates with the final size of the individual because it fuses its epiphyses late in ontogeny (Silver, 1969). Furthermore, the total number of CGMs identified in this bone agrees fairly well with the estimated age of the specimens (Table 2), even in the oldest one, which is aged 8 years and present 7 CGMs (one obscured by haversian systems) in the cross-section. This result provides reliability to the estimation of longevity in wild populations of Asiatic wild ass that are known to live around nine years in the wild (Kaczensky et al., 2015). Horner, De Ricqlès & Padian (1999), in their study of Hypacrosaurus stebingeri, suggested that also the tibia is suitable for skeletochronology. However, the presence of many haversian systems in the tibial cortices of hemionus yearlings (Fig. 1B) indicates that it does not provide accurate skeletochronological results in the Asiatic wild ass. The use of metapodial bones in skeletochronology is still a controversial issue. While Horner, De Ricqlès & Padian (1999) do not recommend it, for perissodactyls, Martínez-Maza et al. (2014) obtained acceptable results in their histological analysis of the fossil species Hipparion concudense. In artiodactyls, however, it does not work because of the ontogenetically late fusion of metatarsus III and IV that deletes growth structures (M Köhler, pers. obs., 2006). Our results show that these bones record a similar total number of BGMs as the femur (Table 2), although the first BGM identified in these skeletal elements seems to be a non-cyclical BGM (Table 2, Figs. 3–6), a fact that must be taken into account when calculating individual age. This information is especially important for studies that comprise a single individual, to not overestimate the results. Moreover, adult metacarpi and metatarsi show a lower number of BGMs than femora within the FLC (Table 2, Fig. 6), which contrasts with the results obtained by Martínez-Maza et al. (2014). The presence of BGMs in the fibrolamellar tissue provides important information about the growth and the timing of key life history traits of the species. Because it is deposited during growth (Huttenlocker, Woodward & Hall, 2013) distance between BGMs has been used to estimate growth rates in extant and extinct mammals (Marín-Moratalla, Jordana & Köhler, 2013; Kolb et al., 2015b). On the other hand, the number of BGMs within the FLC seems to correlate with the time of sexual maturity in artiodactyls (Marín-Moratalla, Jordana & Köhler, 2013; Jordana et al., 2016). Therefore, the results obtained from metapodial bones should be used with caution. Despite these drawbacks, the skeletochronological study of metacarpi and metatarsi still provide valuable individual age estimates because they present a similar total number of BGMs as femora (Table 2). This result is especially interesting for the inference of longevity in fossil species, as these bones are the most abundant remains of equids in paleontological sites.

Regarding bone tissue types, our results show that femora and tibiae present laminar bone (Fig. 1A) while the cortices of metapodial bones are mainly composed of longitudinal POs arranged in circular rows (Fig. 2A) (Francillon-Vieillot et al., 1990). This histological variability, which agrees with previous descriptions of the bone tissue of extant (Enlow & Brown, 1958; Stover et al., 1992) and fossil (Sander & Andrássy, 2006; Martínez-Maza et al., 2014) equid species, is likely related with the specific growth rate and biomechanics of each bone (Horner, De Ricqlès & Padian, 1999; De Margerie, Cubo & Castanet, 2002; De Margerie et al., 2004). On the one hand, the kind of bone matrix is associated with different growth rates (Amprino, 1947; Huttenlocker, Woodward & Hall, 2013) while the arrangement of the vascular canals is commonly related to mechanical forces (De Margerie, Cubo & Castanet, 2002; De Margerie, Cubo & Castanet, 2002). Furthermore, ontogenetic histological changes regarding bone matrix have been noticed in the different limb bones studied. Our study shows a marked change in the proportion of PFB (Fig. 1A) within the FLC matrix in tibiae of subadult kulans. Bone matrix change, along with a modification of the orientation of the vascular canals, has also been observed in femora of E. hemionus (Nacarino-Meneses, Jordana & Köhler, 2016). These histological modifications are likely related to both the changes in loadings (Firth, 2006) and in growth rate (Peters, 1983) that foals experience at the moment of birth.

Amongst all bone tissue types, the occurrence of EFS in vertebrates is a controversial issue. Traditionally, its deposition has been interpreted as the attainment of skeletal maturity (Cormack, 1987; Chinsamy-Turan, 2005; Woodward, Padian & Lee, 2013; Martínez-Maza et al., 2014; Amson et al., 2015; Kolb et al., 2015b) but recent studies have shown that, at least in mammals, it might also be related with the onset of sexual maturity of the species (Klevezal, 1996; Marín-Moratalla, Jordana & Köhler, 2013; Jordana et al., 2016). Growth studies have been shown to provide good estimations of these traits in fossil species (Lee et al., 2013). Our results indicate that the EFS is deposited after epiphyseal fusion in all bones and at a later time in the male than in the female (Table 3, Fig. 7). Actually, in most of the bones analyzed, the time of fusion of both epiphyses agrees with an important drop in the rate of radial growth (inflection point in the growth curves, Fig. 7) and does not match the time of deposition of the EFS. Concretely in the femur, which epiphyses are fused at the age of three (Silver, 1969), the EFS of the wild female is deposited in the fourth year of life while in the wild male it appears at the age of six (Table 3, Fig. 7). In metapodials, the EFS appears after the third year in the female and after the fourth year in the male (Table 3, Fig. 7). These skeletal elements are completely fused at the age of two (Silver, 1969). The correspondence between the pronounced decrease in periosteal growth rate and the age of epiphyseal fusion (Silver, 1969) (Table 3) suggests the decrease in periosteal growth rate to be a good indicator of the end of longitudinal growth in the respective bone. However, the deposition of the EFS some time after growth decline (Fig. 7) indicates that the bone shaft continues growing at minimal rates over some time until full radial growth is achieved (Huttenlocker, Woodward & Hall, 2013). This decoupling between longitudinal and radial growth suggests that inferences of skeletal maturity from the time of deposition of the EFS in equids might be incorrect. However, the presence of the EFS in femora agrees fairly well with the age at first reproduction reported for E. hemionus (Table 3; Kaczensky et al., 2015). In general terms, the femur in mammals presents the longest time of development with the latest epiphyseal fusion (Silver, 1969). Thus, its histological structure should provide the best record of life history events. It is known that although kulans are sexually mature at their second or third year of life (Nowak, 1999), they delay some years its first mating (Kaczensky et al., 2015). Hence, our results provide histological evidence for this well-known behavior in equids (Fielding, 1988; Monfort, Arthur & Wildt, 1994).

Table 3 Age of deposition of the external fundamental system (EFS) in the limb bones of adult kulans and time of several biological traits in equids obtained from the literature.

Age of epiphyseal fusion (Silver, 1969) is indicated for the closely related species Equus caballus while age at sexual maturity (Nowak, 1999) and age at first reproduction (Kaczensky et al., 2015) is reported for Equus hemionus. All data are expressed in years.

	EFS	Epiphyseal fusion			
	F	T	Mc	Mt	F	T	Mc	Mt	Sexual maturity	Age at first reproduction	
Female	4	−	3	3	3–3.5	3–3.5	1.25–1.5	1.3–1.6	2	3	
Male	6	–	4	4	3–3.5	3–3.5	1.25–1.5	1.3–1.6	3	5	
Notes.

F femur

T tibia

Mc metacarpus

Mt metatarsus

Finally, the growth analysis has also revealed a high inter-individual variability in size (Fig. 7) that should be taken into account when retrocalculating lost CGMs. Our results, although obtained from a relatively small sample size, show different femoral growth tendencies between wild and captive individuals (Fig. 7A) and a higher growth rate in captive exemplars than in wild ones during the first year of life (Figs. 7E–7H). These differences, that reflect the influence of the habitat in the life history of the species, have previously been reported for mammals (Marín-Moratalla, Jordana & Köhler, 2013) and alligators (Woodward, Horner & Farlow, 2014) and are related with the constant food supply and care that captive animals experience during their life (Asa, 2010). To obtain the most accurate data, we propose to study wild animals when possible to avoid overestimation of growth rates for the species under study.

Conclusions

Our study analyzes the histological variation between different limb bones of the Asiatic wild ass. Our research provides evidence that the femur is the most reliable bone for skeletochronological studies in equids, although metapodial bones also provide good individual age estimations. The use of tibiae, however, is not recommended for this group due to the high presence of secondary osteons observed in early ontogenetic stages. Furthermore, all bones present histological changes regarding the proportions of bone matrix components and / or the arrangement of vascular canals in the course of ontogeny. Finally, the presence of an EFS in the outermost cortex of adult femora is likely related to the reproductive maturity of the species (first reproduction) than to skeletal maturity. Skeletal maturity, however, is recorded in growth curves as a significant drop in periosteal growth rate.

We are grateful to Thomas Kaiser for loans of the collection of the Zoological Institute of Hamburg University (Hamburg, Germany) and to Renate Schafberg for permission to cut the adult kulan bones from the collections housed at Museum of Domesticated Animals of the Martin-Luther-University Halle-Wittenberg (Halle, Saale, Germany). We are indebt to Gemma Prats-Muñoz and Luis Gordon for preparing the thin sections of the study. Finally, we acknowledge William Jungers for handling the article as editor of PeerJ and Clara Stefen, Jorge Cubo and Tim Bromage for their useful reviews, which greatly improve an earlier version of the manuscript.

Additional Information and Declarations

Competing Interests

Author Contributions

Data Availability

The authors declare there are no competing interests.

Carmen Nacarino-Meneses conceived and designed the experiments, performed the experiments, analyzed the data, wrote the paper, prepared figures and/or tables, reviewed drafts of the paper.

Xavier Jordana analyzed the data, reviewed drafts of the paper.

Meike Köhler conceived and designed the experiments, analyzed the data, contributed reagents/materials/analysis tools, reviewed drafts of the paper.

The following information was supplied regarding data availability:

The raw data is included in the results.

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
