# Peer review of "Histological variability in the limb bones of the Asiatic wild ass and its significance for life history inferences"

_PeerJ, doi:10.7717/peerj.2580_

## Round 0.1 · original submission · Minor Revisions

With 3 reviews in hand, all indicating "minor revisions", I am happy to inform you that your paper should be acceptable for publication in PeerJ pending attention to the reviewers' various recommendations and requests for clarification. Reviewer #2 raises the most substantive issue that deserves special consideration as you prepare your revised ms. (which I hope will not need to be sent out for re-review).

·

Basic reporting

The paper is worth to be publisched as it presents new results and not many mammals are studied in detail concerning bone histology and skerlotochronology yet. Also the idea to compare the results in different bones is worth while and not seen to often. It is generally well written, understandable (with few exeptions see below) and uses relevant and recent references.
However, I would suggest few modifications or additions and have some questions:
It is not quite clear to me how the word "longevity" is used in this paper. It seems to be used rather in the meaning of "individual age of the studies specimens" than "longevitiy of the species" (as I understand the assumend or known highest age individuals of a species can live up to. To make assumptions on the latter would be rather possible on the basis of the limited material studied. This is another drawback: the sample size should be larger to corroborate the conclusions drawn in particular in respect to sex as there are several For the non-specialist on bone histology (like eoclogists interested in age determination or so) a scheme explaining the structures (long bone, where are lines, how do the different bone types discussed look) in bone discussed would be very helpful to follow the paper and in particular the pictures.
Specific omments:
p.6 l 177 - wording "indicates" seems odd
p 7 l 205/06 it is not quite clear why this line is interpreted as non cyclical? Is the position of the line the main point or being produced in winter?
The same comment holds for p. 8 l 213-216.
Is a comparison to the lines counted in the tooth cement (if known from the museum records in halle) useful?
p 10 l 268 place "for the fist time" at the end of the sentence
Some of the statements in the discussion are very general and indicate very general conclusions hard to make on the basis of the small sample size, even if they are not wrong they imply more than there is. E.g. l 274" Our thorough stuy of ..." or l 282 "The results obtained .. will serve as basis for inference of life history ...
p10 l 288 and following: I am not sure that the study of bone histoloy in this particular species here provided so new data on life history traits, as reproduction time, growth, time of reproduciton etc. are knwon. Bone histology can help to elucidate them for specimens found dead and to sketh the individual lives better - I guess that should be stressed.
How does number of lines in femur and teeth math (if knwon)?
p 11 l 310"register" wording seems odd
l 315 follwong "The presence of BGMs ..." a brief explanation why should be included here for clarity even though a reference is given.
l 319 - here again one of my main comment: the use of "longevity" indicates a too generalized conclusion.
P11/12 paragraph on variability of bone histology is difficult to follow without some more specific comments on why in ontongenetic and biomechnic context the one or other would be better/more adapted or what. It would help the reader to include some information even though literature is given.
p 13 paragraph starting line 366 - the sample size is small so the conclusions should reflect this and indicate more caution
table 1 - if known the number of cement lines should be listed as well
table 3 heading or table are years or months given? and dots instead of commas for ages given
figures
maybe some show the details more clearly in black and white
indicate what type of microscopy was used: normal transmission light or others (e.g. phase contrast or so)
Fig. 1 B ) lable the harvesian systems, I am not sure I see it right, C) add lable PFB like in fig 1A D) what is in between the "LB" Zones and what is the difference - it is hard to see
Fig 2 A are the osteons directed radially? D) what is below the EFS?
Fig 3 the pictures are very small and the arrows as well as growth marks hard to see
Fig 4 C) why 1 arrow isn't there a second line above the one marked?
D) why the lower arrow? if this is a line why is there not another above (also in the dark zone)
Maybe it helps to describe briefly in the legend what line and why the light or dark line is counted in fig 4 and 5
fig 7 labeling of the axis is too small, try to used a uniform type of symbol (e.g. full or unfilled) for females versus males and unsexed animal and indicate wild and captive animals with color or type of line - that would help to understand the figure more rapidly
lable which bone in the figure and try to indicate in the figure or legend what you try to sy on page 12 line 344 onwards - it was not all clear to me directly.

Experimental design

good,

Validity of the findings

sample size could be larger to corroborate the findings, but ok

·

Basic reporting

No comments

Experimental design

Although most paleohistology papers are based on transverse sections, Stein and Prondvai (2014) showed that longitudinal sections must be analyzed as well. This is because, under polarized light, dark regions on a transverse section can correspond to either parallel fibered bone cut transversally or to woven bone. For this reason, longitudinal sections must absolutely be used. Considering that you identified these kinds of bone tissue in your study, please explain why longitudinal sections were not used.

Validity of the findings

Findings shown in the Ms coauthored by Nacarino-Meneses et al. are based upon currently accepted and widely used paleohistology methods. Therefore these findings have been obtained using the current state of the art. As I reviewer I suggest, however, to discuss in the Ms a methodological point that is potentially problematic:
Authors state that "The number of CGMs within a bone cortex allows
researchers to calculate important life history traits such as longevity". However, considering that mammals have a determinate (asymptotic) growth, and that lines of arrested growth are difficult to identify at the periphery (because it is formed by parallel fibered to lamellar bone tissue), to estimate longevity using this approach is very approximative (of low accuracy) at best. Are longevity inferences based on this approach reliable? Please explain.

Additional comments

Dear authors
The Ms entitled "Histological variability in the limb bones of the Asiatic wild ass and its significance for life history inferences " is an interesting contribution that merits publication provided that you deal with the points raised above on methodological issues.
Yours sincerely
Jorge Cubo

·

Basic reporting

No Comments

Experimental design

No Comments

Validity of the findings

No Comments

Additional comments

It is an excellent paper in all respects. I have only minor comments in the attached PDF

---

## Round 0.2 · accepted · Accept

Your detailed response to the 3 reviewers is greatly appreciated. Your attention to the various suggestions has served to improve your manuscript and clarify your important findings. I believe your paper is now acceptable for acceptance in PeerJ. If possible, your final version might benefit from a quick edit by a native English speaker, but this suggestion is not obligatory.